# Precision Nutrition Unveiled: Gene–Nutrient Interactions, Microbiota Dynamics, and Lifestyle Factors in Obesity Management

**DOI:** 10.3390/nu16050581

**Published:** 2024-02-20

**Authors:** Samy Mansour, Saif M. I. Alkhaaldi, Ashwin F. Sammanasunathan, Saleh Ibrahim, Joviana Farhat, Basem Al-Omari

**Affiliations:** 1College of Medicine and Health Sciences, Khalifa University of Science and Technology, Abu Dhabi P.O. Box 127788, United Arab Emirates; 100062686@ku.ac.ae (S.M.); ashwin33377@gmail.com (A.F.S.);; 2Institute of Experimental Dermatology, University of Lübeck, Ratzeburger Allee 160, 23538 Lübeck, Germany; 3Department of Public Health and Epidemiology, College of Medicine and Health Sciences, Khalifa University of Science and Technology, Abu Dhabi P.O. Box 127788, United Arab Emirates

**Keywords:** precision nutrition, obesity, translational medicine, genetics, microbiome, lifestyle factors

## Abstract

Background: Obesity is a complex metabolic disorder that is associated with several diseases. Recently, precision nutrition (PN) has emerged as a tailored approach to provide individualised dietary recommendations. Aim: This review discusses the major intrinsic and extrinsic components considered when applying PN during the management of obesity and common associated chronic conditions. Results: The review identified three main PN components: gene–nutrient interactions, intestinal microbiota, and lifestyle factors. Genetic makeup significantly contributes to inter-individual variations in dietary behaviours, with advanced genome sequencing and population genetics aiding in detecting gene variants associated with obesity. Additionally, PN-based host-microbiota evaluation emerges as an advanced therapeutic tool, impacting disease control and prevention. The gut microbiome’s composition regulates diverse responses to nutritional recommendations. Several studies highlight PN’s effectiveness in improving diet quality and enhancing adherence to physical activity among obese patients. PN is a key strategy for addressing obesity-related risk factors, encompassing dietary patterns, body weight, fat, blood lipids, glucose levels, and insulin resistance. Conclusion: PN stands out as a feasible tool for effectively managing obesity, considering its ability to integrate genetic and lifestyle factors. The application of PN-based approaches not only improves current obesity conditions but also holds promise for preventing obesity and its associated complications in the long term.

## 1. Introduction

Obesity is a complex metabolic disorder that can present with cardiovascular diseases (CVD), type 2 diabetes mellitus (T2DM), non-alcoholic fatty liver disease (NAFLD), dyslipidaemias, and cancer [1,2,3]. According to the World Obesity Atlas 2023, 38% of the global population is currently either overweight or obese [4]. This progression of obesity has been primarily associated with the consumption of an unbalanced diet rich in fat and fructose for a long period while adopting a sedentary lifestyle [5,6]. Therefore, weight control is vital to prevent diseases. It is suggested that controlling weight is influenced by the sources and quality of food rather than the quantities consumed in the diet [7]. It is also suggested that the individual’s genetic and epigenetic interactions with dietary intake and physical activity are linked to an increased risk of developing obesity [8,9]. In case of intrinsic disruptions, a high intake of saturated fat or refined carbohydrates results in dysregulation of the central metabolic pathways and increased weight [10,11]. Additionally, the gut microbiota and its interactions with genes and diet can modify the risk of developing obesity [12].

The multifactorial nature of obesity has led to the ongoing application of precision public health approaches to enhance the understanding of the interplay between individuals’ intrinsic components and environmental factors [13]. These precision approaches have been implemented to ameliorate patients’ health and quality of life [14]. In recent years, precision nutrition (PN) was introduced as one of these approaches to provide customised dietary recommendations for individuals based on their genetic profile, microbiota, physical activity, and lifestyle [15,16]. In particular, PN has been applied in metabolic conditions such as obesity to provide individualised metabolic care and clinical nutrition [17]. PN focuses on biological biomarkers characterising each individual to apply more effective and personalised nutritional guidelines [18]. This enables patient subgroups to obtain customised nutritional instructions rather than general recommendations to improve their treatment outcomes [19,20]. In the long term, PN has been associated with a promising potential to extend patients’ health span and limit the burden of healthcare costs [21]. 

It is worth mentioning that the concept of PN requires a higher degree of certainty compared to other precision approaches [22]. PN provides sufficient information regarding complex correlations between biological factors, food intake, and phenotype to share individualised nutritional advice [23]. This prioritises the application of PN in case of elevated genetic susceptibilities to specific diseases, such as obesity [22]. This review will discuss the major intrinsic and extrinsic components considered when applying PN. This review will also focus on the practical contribution of PN during obesity management and associated common chronic conditions. 

## 2. The Phenotypic and Genotypic Components of PN

In the last couple of years, multiple approaches have been designed to focus on nutrition and food science technology [1]. These advanced methodologies are based on understanding the individual variability in response to foods to provide personalised nutritional recommendations specific to patient subgroups [2]. PN is one of the promising methods that have been used for approaching the variation in individuals’ responses to diet, nutrients, metabolic activity, and treatment outcomes [3]. These variations have been linked primarily to the composition of the gut microbiome, genetics/metabolic profile, and social and lifestyle habits specific to each individual [4]. The gut microbiome has been classified as one of the factors that can predict individuals’ responses to diet and develop an appropriate model for PN [5]. Understanding the variation in genetic and metabolic profiles can also help in providing specific dietary advice for individuals or population subgroups in the form of PN [6]. PN can optimise the dietary response and health by considering the variations in individuals’ social status and lifestyle habits [7]. 

PN allows a better understanding of the inter-individual differences that are directly correlated to patients’ unique intrinsic factors, including the microbiome as well as the genetic and metabolic profile [8,9,10,11]. Other factors including patients’ health status, physical activity, and dietary pattern, and psychosocial and socioeconomic characteristics can also extrinsically affect the response to dietary behaviours [12,13,14,15], as shown in Figure 1 and explained in the following sections.

### 2.1. Gut Microbiota

The gut microbiota refers to the intestinal tract microorganisms responsible for the generation of metabolites, stabilisation of homeostasis, and maintenance of adequate immune responses [16,17]. The intestinal microbiota is also connected with the brain axis to allow the exchange of information across the hypothalamus, pituitary, and adrenal glands. This interconnection is responsible for activating the dual hunger–satiety circuit and the dopamine reward path, producing energy and acquiring food from the environment [18]. The axis between the brain and intestines can be influenced by the host’s genetic composition, level of stress, negative emotions, and diet type [19]. The microbiota can also regulate the pathogenesis, progression, and management of diseases [20,21]. The effectiveness of these functions relies on both the quantity and quality of the microbiota, as well as its metabolic potential [22]. This shows that the characteristics of the gut microbiota can significantly vary across individuals based on their genetic profile, lifestyle, and habits [23]. In practice, the gut microbiota is recognised as a key determinant in predicting how individuals respond to particular dietary components [24]. Consequently, the direct evaluation of host–microbiota interactions constitutes an advanced therapeutic tool during disease control and prevention stages [25]. This highlights the necessity of evaluating individuals’ microbiota to structure precision diets and interventions required for optimal health [26].

In daily life, the type of consumed diet has been shown to influence the microbiome’s composition. For example, diets rich in animal-based nutrients can stimulate the release of bile-resistant species, while plant-based foods are associated with a higher level of plant polysaccharide-fermenting species [27]. A limited 24 h consumption of carbohydrates can also decrease the production of bacteria responsible for destroying food fibres [28]. In obesity and weight gain cases, the microbiota plays a crucial role in monitoring energy use and the formation of gut metabolites [29]. Therefore, individuals eating an unhealthy diet and gaining weight for a prolonged period of more than ten years have limited intestinal microbiota diversity [30]. Other findings relate the disruption in the microbiota’s composition to some inherited and non-modifiable individual characteristics, such as ethnicity and geographical setting [31]. 

Another factor that can also interfere with the variety and stability of gut microbiota is the consumption of sugar substitutes [32]. It was found that the administration of sucralose for 12 weeks was associated with lower levels of anaerobes, *bifidobacteria*, *lactobacilli*, *Bacteroides*, *clostridia*, and total aerobic bacteria [33]. Prolonged sucralose consumption in mice caused a high release of bacterial pro-inflammatory genes and disruption in faecal metabolites [34]. In turn, the limited utilisation of emulsifiers as food additives showed a reduced microbial diversity [35]. The consumption of a fermentable oligosaccharides-, disaccharides-, monosaccharides-, and polyols (active b)-rich diet can also alter the gut microbiota [36]. This type of diet has been shown to minimise the risk of insulin resistance in healthy overweight and obese patients [37,38].

Animal and in vitro studies found that gluten-free bread intake lessens the microbiota dysbiosis usually occurring in gluten sensitivity or coeliac disease cases [39,40]. Following a four-week gluten-free diet (GFD), individuals presented with different metabolic profiles and subsequent changes in gut microbiota [41]. In healthy subjects, decreased levels of Bifidobacterium, *Clostridium lituseburense*, *Faecalibacterium prausnitzii*, and *Lactobacillus*, and higher *Enterobacteriaceae* and *Escherichia coli* counts following GFD, were reported [42]. There is some emphasis on the potential of a low-gluten diet to induce moderate changes in the intestinal microbiome, reduce fasting and postprandial hydrogen exhalation, and improve self-reported bloating in healthy individuals [43]. Despite its advantages, GFD can be associated with a higher risk of heart disease due to a possible reduction in whole grains’ consumption [44]. 

This may suggest the impact of dietary habits on gut microbiota and confirm the crucial role of the microbiome as a major contributing factor to PN (Figure 2). Therefore, the variation between individuals and populations can affect the overall response to diet in terms of meals’ digestion, nutritional benefits, and personalisation.

### 2.2. Genetics and Metabolic Profile

The genetic makeup is another important factor that may contribute to the inter-individual variation in dietary behaviours [45]. Some of the recent advances in the field of genomics assisted researchers in better understanding the role of genetic variant sites and functions in the development of chronic conditions. This also contributed to predicting the risk of chronic diseases and personalising their prevention and treatment plans [46]. Accordingly, individuals may present with genetic variations, known as polymorphisms, which can lead to differences in the metabolic processing of nutrients within the same population [47]. For example, the presence of a single-nucleotide polymorphism (SNP) in intron 1 of the cytochrome P450 enzyme *CYP1A2* gene was linked to a variation in caffeine metabolism [48,49]. This may account for the high inter-individual variability found in caffeine intrinsic concentrations. In addition, individuals with the CC genotype of a SNP were more likely to gain weight when eating a high-saturated-fat diet (around 10% higher BMI), whereas those with the TT genotype were not associated with this complication [50,51]. 

The mutual mapping of the individuals’ genetic and metabolic profiles constitutes an important tool for assessing the body’s response to different nutrients and tailoring a personalised diet [52]. For example, a protein- and fibre-rich diet may benefit individuals suffering from low-insulin sensitivity, while a high monounsaturated fatty acids-based diet may benefit patients with insulin resistance [53]. Interestingly, when measuring the metabolic response to high and low glycaemic index meals, it was found that certain subjects had a variance in their glucose and insulin responses to the same standardised index meals [54]. This further strengthens the importance of taking into consideration the inter-individual alterations in metabolic profile while tailoring the diet to produce better health outcomes. In parallel, biological components, including proteins, metabolites, microbiota, and epigenetic markers, helped in understanding the possible association between the individuals’ physiological mechanisms and their susceptibility to chronic diseases [55].

### 2.3. Psychosocial and Socioeconomic Status

Obesity has been correlated with an increased risk of psychosocial burden [56]. In particular, obese individuals can suffer from mood, self-esteem, and body image-related issues [57]. It was suggested that depressive symptoms are associated with altered eating behaviours and increased food and beverage caloric intake [58]. This is likely due to the food’s ability to activate brain reward circuits that lead to the release of dopamine [59]. Patients with eating disorders also reported difficulties in controlling the frequency of their eating, portion sizes, or extreme eating behaviour [60]. This can be further associated with disordered eating leading to uncontrolled weight gain. In such cases, the psychosocial status of individuals should be considered as a crucial factor when tailoring a specific nutritional plan [61]. The implementation of a long-term weight loss plan should also be considered to allow a gradual improvement in an individual’s psychological distress [62].

Socioeconomic status can also be directly correlated with psychological status [63], as individuals with a high socioeconomic status (SES) seem to follow healthier food habits [64]. Individuals with a low SES are prone to a poorer health status, as they are unable to comply with the required nutritional recommendations or dietary guidelines; in turn, they are more likely to experience unhealthy conditions [59,63]. Recent evidence suggested that a higher percentage of low SES households had unhealthy eating habits, such as consumption of fast foods, while high SES households had healthier eating patterns [65]. This could be due to the lack of purchasing power in those with a low SES, which can lead to the consumption of cheaper and lower-quality ingredients, causing nutritional deficiencies [66]. This confirms the importance of SES consideration when tailoring an individualised diet to improve the nutritive quality. Moreover, both social inequity and diet quality, in conjunction with healthy dietary behaviours, constitute a crucial and active public health concern.

## 3. PN Use in the Management of Obesity

Obesity is a complex non-communicable disease, which is influenced by both environmental and hereditary factors, representing a relevant target for PN [67]. Recently, studies have shown significant correlations between individuals’ intrinsic components and the extrinsic factors that can directly affect their lifestyle and habits [67]. In this case, PN can help in simultaneously evaluating individuals’ genes, metabolic markers, microbial species, environmental elements (sociodemographic and physical activity), and obesity phenotypic traits (body weight, body mass index, waist circumference, and central and regional adiposity).

### 3.1. Genetic Basis of Obesity

The progression of obesity has been correlated with multiple genetic factors that can affect the interaction of macronutrients with the individual’s genotype [68,69]. Recently, advanced human genome sequencing and applied population genetics studies have been used as a main tool for detecting the gene variants associated with obesity and its related traits [69]. SNPs have been known as one of the main types of genetic variants that are associated with obesity [70]. For example, insulin-like growth factor, dioxygenase enzyme, melanocortin receptor, and apolipoprotein present SNPs associated with an increased risk of obesity [1,2,16,17,18,71]. In parallel, the conduction of a Genome-Wide Association Study (GWAS) helped in detecting more than 140 obesity-related SNPs [72], while another one revealed about 300 SNPs in total [73]. Despite their discovery, the impact of these SNPs on BMI is still modestly rated, and further investigations are needed for evaluating the influence of genetics on the development of obesity [70,73]. This cannot be relevant to Prader-Willy syndrome cases or genes related to leptin and melanocortin signalling, which have a higher influence on BMI [74]. 

In some cases, genes are integrated with carbohydrate and lipid metabolism [75], and in others, genes are responsible for activating the proteins of carbohydrates and lipid taste receptors [76]. Some of the genes participate in encoding the lipid transporters of proteins or digestive enzymes present in starch and milk [77], and some other genes are responsible for using and storing energy, food reward, and gut regulatory processes [78,79]. For example, the intake of dietary fats and carbohydrates was found to be associated with the SNP “rs1761667” and “rs35874116” on the cluster of differentiation 36 (CD36) protein and taste 1 receptor member 2 (TAS1R2) gene, respectively [18]. Moreover, “rs1799883” SNP of the fatty acid binding protein 2 (FABP2) gene was found to be associated with hypertriglyceridemia, while “rs9939609” SNP of the fat mass and obesity-associated (FTO) gene was correlated with an increased risk of body fat accumulation [18]. In parallel, the “rs1800497” SNP of the dopamine receptor D2 (DRD2) gene was seen to interlink with the brain–gut microbiota axis and stimulate dysbiosis, negative emotions, and obesity [18]. Despite their unsatisfactory effects, some genetic variations, such as a high AMY1 copy number, were seen to protect the body against obesity [80]. Other genetic variants, polymorphisms, and SNPs are presented in Table 1.

It is worth mentioning that epigenetic changes linked to external factors can change genetic activity and express the obese phenotype [93,94]. These epigenetic modifications were mainly documented following the practice of a dietary plan, physical activity, and surgeries [13]. Therefore, advanced gene-based technologies have been implemented to personalise dietary recommendations based on individuals’ genetic profiles [95]. Nutrigenomics studies reported how individuals’ genetic variations influence their responses to nutrients and how diet, in turn, affects gene expression [96]. Limited data supported the superiority of this technology regarding weight loss results in comparison to standardised care [97,98]. Currently, the increased application of pharmacogenomics to evaluate therapeutic responses to pharmaceutical compounds is associated with a marked clinical relevance [99]. This advanced technique allows the mapping of genetic variants that can influence the response to weight loss treatment [93]. However, pharmacogenomics application is still limited due to financial issues. In recent studies, the discovery of next-generation probiotics has been linked to several health benefits [100]. The clinical use of these probiotics is also limited due to several reasons, such as safety, efficacy, and cost [101,102,103,104]. In some cases, the transplantation of faecal matter has been initialised for the treatment of obesity and other metabolic disorders. Its clinical application remains limited due to the variation in the obtained findings [105].

### 3.2. Weight Management

The consumption of specific types of food can help prevent the development of obesity [106]. According to epidemiological research, consuming dairy products and vegetarian protein sources can protect against obesity, unlike consuming large amounts of meat, which is correlated with a greater risk of weight gain [107,108,109]. This was supported by a Chinese study, which viewed that consuming large quantities of fatty food can increase the chance of weight gain and obesity [110]. Therefore, poor diet quality was strongly correlated with a greater risk of weight gain despite gender differences [111]. The increased caloric density of high-fat foods also promotes low-satiety effects, especially when consumed in large quantities [112]. This can emphasise the need for a more passive focus on selecting the appropriate intervention for dietary self-monitoring adherence [113]. 

In practice, the majority of obese women valued the use of weight management services and advice, despite the limited practical discussions and application of these approaches [114]. The use of individualised information while providing nutritional advice helped in sustaining changes in healthy dietary behaviours [115]. A UK-based study showed that applying PN advice through mobile applications was beneficial in improving diet quality and individuals’ engagement in dietary habits, in comparison to the general population [116]. These findings highlighted the potential of PN to improve individuals’ adherence to dietary habits as well as improve weight and glucose management for a long period [117,118,119]. In the long term, the benefits of PN use were associated with a decrease in total fat intake, in addition to compliance with nutritional advice [120]. 

In daily life, limited physical activity due to a sedentary lifestyle, high screen time, processed meats, physical education, and transportation constitute a direct risk factor for obesity [121]. For example, individuals exercising minimal physical effort have a greater chance of alleviating the risk of type 2 diabetes by 26% compared to unenergetic ones [122]. When applied to physical activity, PN is still inconsistently modifying behavioural changes, such as the ones documented in dietary patterns and diet quality [123,124,125]. The value of personalising dietary advice for modifying physical activity levels was not found to be superior to the conventional guidelines when the physical activity was objectively assessed [126,127]. In some cases, PN was seen to significantly enhance individuals’ exercise frequency when they were informed about their genetic testing results [128]. 

In summary, the obtained findings can illustrate crucial inputs about PN application in practice and its impact on changing individuals’ diet quality and physical activity, highlighting the need for further research.

### 3.3. Intestinal Bacterial Flora

The different responses to nutritional recommendations can be regulated by the composition of the gut microbiome for each individual [129,130]. It has been evidenced that the intestinal microbiota is intrinsically affected by the overall health, including obesity risk [131]. Physiologically, the gut microbiota utilises energy from the diet and interacts with the host genes that regulate the expansion and storage of energy [132]. Therefore, live bacteria (probiotics), nondigestible or limited digestible food constituents, such as oligosaccharides (prebiotics), or both (synbiotics), or even faecal transplants have been used as an emerging tool to restore the intestinal microbiota and treat or prevent obesity [133,134,135,136]. This may suggest the key role of the gut microbiota while applying the PN criteria to facilitate weight loss in obese individuals [137]. A human study reported that obese individuals had more Firmicutes and nearly 90% fewer Bacteroidetes than lean individuals [138]. It was observed that a healthy weight and good metabolic health were seen in patients with bacteria of the genus *Oscillospira*, while *Collinsella aerofaciens* bacteria were more frequently documented in obese individuals [139]. Other anatomical and physiological changes can also occur in the gut microbiota following the performance of bariatric surgery. Despite its documented benefits regarding weight loss and glycaemic control, a recent study verified the alterations in microbial diversity and composition three months following bariatric surgery [140]. More specifically, a high level of Proteobacteria and Bacteroidetes and a low Firmicutes concentration were reported [141]. In the long term, microbiota species were still maintained, indicating that bariatric surgery could achieve a fast and prolonged modification in the patient’s gut microbiota [141]. The transplantation of faecal microbiota has been reported to reduce body fat accumulation two weeks post-procedure in germ-free mice who had their Roux-en-Y Gastric Bypass (RYGB) or sleeve gastrectomy (SG; 46% and 26%, respectively) [140]. Thus, the inter-individual changes in gut microbiota should always be considered when structuring a nutritional plan, since it can directly influence metabolism, resulting in either weight loss or gain [142]. This was supported by a study validating that individuals’ gut microbiota can be used to design personalised diets for glucose homeostasis [143].

## 4. Obesity-Related Complications

The complex nature of obesity is not only related to its pathophysiological features and symptoms’ severity but also to its associated chronic complications, including type 2 diabetes, cardiovascular diseases, and cancer [144,145,146]. This emphasises the value of PN in alleviating the severity of obesity and the risk of other chronic diseases that may be associated with a poor prognosis and low quality of life [147,148].

### 4.1. Diabetes

Type 2 diabetes (T2D) is a fast-spreading metabolic condition and is rated as the 7th leading cause of death in the United States [149]. Obesity was found to be strongly associated with the progression of T2D, with a seven times greater risk of developing T2D in obese patients than normal weight individuals [150,151]. The maintenance of a balanced metabolic homeostasis is required to effectively manage the condition [152]. This can be achieved by applying PN to ensure an appropriate nutritional consumption of protein, fat, carbohydrates, vitamins, and mineral substances [153,154]. In parallel, specific analytical determinants have been analysed in plasma and urine samples to objectively quantify an individual’s nutritional status [155]. These biomarkers, obtained from biological samples, can provide a more realistic evaluation of nutritional status than dietary intake [156]. Figure 3 provides some examples of evidence-based nutritional biomarkers in plasma and urine samples.

The low adherence to the Mediterranean diet can cause a higher risk of T2D progression [157]. In diabetic patients, diets low in sugar-sweetened drinks and processed meats, and high in fruits and vegetables, whole grains, and nuts, can improve the glycaemic index and lipid abnormalities [158]. This type of diet can also reduce the risk of acquired diabetes in people susceptible to disease development [158]. Thus, the consumption of specific nutrients and avoidance of others is a key factor in promoting insulin sensitivity for the prevention and treatment of T2D [159]. In pregnant women, the TCF7L2 gene and its related TCF7L2 rs790314 C>T polymorphism are correlated with a higher risk of gestational diabetes mellitus (GDM) [160]. The TCF7L2 gene rs790314 TT genotype has a higher susceptibility to diabetes, which may be alleviated if a Mediterranean diet is followed to reduce the genetic risk [160]. The intestinal microbial flora also differs, with lower Firmicutes seen in diabetic patients when compared to the non-diabetic healthy controls [161,162]. However, further confirmation by causal, quantitative, and/or appropriate mechanistic analysis is worth pursuing to confirm this correlation and association phenomena. More specifically, *Prevotellacopri* (*P. Copri*) and *Bacteroidesvulgatus* species were seen to stimulate insulin resistance by activating the branched-chain amino acids (BCAAs) biosynthesis [163]. Intrinsically, BCAAs signal the mammalian target of the rapamycin (mTOR) complex, leading to phosphorylation of p70S6 serine kinase (S6K1), which inactivates the insulin receptor substrate responsible for promoting insulin resistance [164]. The ingestion of barley kernel bread was seen to improve the postprandial glucose and insulin resistance in patients with T2DM due to the higher Prevotella/Bacteroides ratio [165]. This result supports the characterisation of gut microbiota for each patient with obesity and T2DM to better predict the host and microbiota response to dietary interventions [166].

The application of PN along with its intrinsic and extrinsic components can help in improving the well-being of obese patients and avoiding the occurrence of other serious complications, such as diabetes.

### 4.2. Cardiovascular Diseases

Cardiovascular disease (CVD) is a general term including a class of disorders that involve the heart or blood vessels [167]. CVD often results from a combination of factors, including genetics, lifestyle habits, and environmental factors, which can lead to complications such as heart attacks and strokes [168]. In recent years, heart disease has been characterised as the leading factor behind the elevated cases of morbidity and mortality [169]. 

Heart failure is a clinical syndrome affecting over 60 million people and is associated with the inability of the heart to pump or fill blood [170]. PN can be applied for the prevention of heart failure and even in treatment stages to improve patients’ prognosis and quality of life [171]. A recent nutritional study focusing on the Dietary Approaches to Stop Hypertension (DASH) diet discussed the importance of prioritising fruits and vegetables, whole grains, fish, low-fat dairy, poultry, and lean meat intake [171]. The consumption of seeds, legumes, and nuts while limiting the overuse of oils and fats is also recommended. In turn, it was found that participants positioned in the top quartile of the DASH diet score had a 37% lower risk of heart failure [172]. The adherence to a Mediterranean diet showed a protective effect against myocardial infarction and a strengthened triglycerides-lowering effect [157,173]. This may be due to the high amount of fibres enriched in this type of diet contributing to the alleviation of the risk of heart disease through high anti-inflammatory potential [174,175]. Further reports showed that dietary fibre intake can help in alleviating the risk of cardiovascular diseases, mainly atherosclerosis [169]. For example, peripheral artery disease (PAD), which is strongly interlinked with inflammation, has been evaluated based on the dietary inflammatory index (DII) [176]. Consequently, a higher DII score has been associated with poor health and increased risks of obesity, cancer, cardiovascular disease (CVD), chronic obstructive pulmonary disease (COPD), depression, metabolic syndrome, T2D, and kidney stones [177,178,179,180,181,182,183]. This confirms the value of a diet rich in fibre, folate, and vitamins A, B6, C, and E in alleviating the prevalence of PAD [184]. It was also reported that a daily intake of moderate amounts of olive oil can have a protective effect on atherosclerosis due to its micro-constituents’ anti-atherogenic potential [185]. These findings highlight the crucial role of nutritional recommendations to help individuals prioritise the consumption of beneficial foods and, in turn, reduce the prevalence of cardiovascular diseases.

### 4.3. Cancer

Cancer is a disease characterised by the malignant proliferation of cells that have managed to elude endogenous regulatory systems [186]. In 2016, The International Agency for Research on Cancer (IARC) reported that weight control is an important factor in alleviating cancer risk [187]. Evidence obtained from epidemiologic trials, mechanistic studies, and animal models also supported the association between obesity and different types of cancer in terms of risk and mortality [188]. Physiologically, the release of adipokines normally documented in obese patients was associated with the progression of obesity-related cancers; however, this causal association is still subject to preliminary analysis [189]. This finding was not supported in pancreatic cancer, renal cell carcinoma (RCC), ovarian cancer, and endometrial cancer. Moreover, leptin, soluble leptin receptor (sOB-R), and plasminogen activator inhibitor 1 (PAI-1) were also similarly unrelated to the risk of obesity-related cancers [190]. 

In daily life, there is a large variety of bioactive phytochemicals found in food that can regulate tumour development, metastasis, and progression [191]. For example, green tea flavonoid epigallocatechin-gallate (EGCG) has been demonstrated to reduce tumour growth by increasing apoptosis and decreasing the proliferation of tumour cells [191]. This substance makes human colon cancer cells more susceptible to 5-fluorouracil, improves prognosis, amplifies the effect of adjuvant therapy, and lowers the probability of tumour relapse [192]. Fruits and vegetables were also found to have a protective effect against cancer since insufficient ingestion of these nutrients can increase cancer risk [193]. Indeed, a systematic review of 110 high-quality studies found that fruit and vegetable consumption had a protective effect against lung, breast, and colorectal cancer [194]. Lastly, meta-analyses of case-control and cohort studies showed that whole-grain consumption causes a decreased risk of both site-specific and total cancer [195]. In summary, the consumption of specific nutrients reduces the risk of cancer through numerous mechanisms, such as the downregulation of tumour proliferation or the promotion of cell death in cancerous cells [196].

## 5. Conclusions

PN stands out as a crucial tool for enhancing the management of various obesity-related risk factors, including dietary patterns, physical activity, body weight, and metabolic markers. Supported by omics technology, PN delves into individual genetic, biomarker, and gut microbiota variations, revealing personalised responses to dietary inputs. PN’s promise extends beyond weight management, positioning it as a key asset for regularly evaluating overall patient health, especially in the presence of concurrent illnesses. In current practice, the lack of personalised dietary data may still restrict the robust application of PN. The high technological cost along with the deficient number of professionals able to analyse and evaluate this customised information can impose additional challenges when applying PN. Additionally, the association of obesity and its related complications, such as T2D with multiple polymorphic genetic variations, can make dietary interventions even more complex. Consequently, the appropriate application of PN will help in customising dietary plans to ensure patients’ adherence to the treatment and prevention of additional adverse conditions. 

## Figures and Tables

**Figure 1 nutrients-16-00581-f001:**
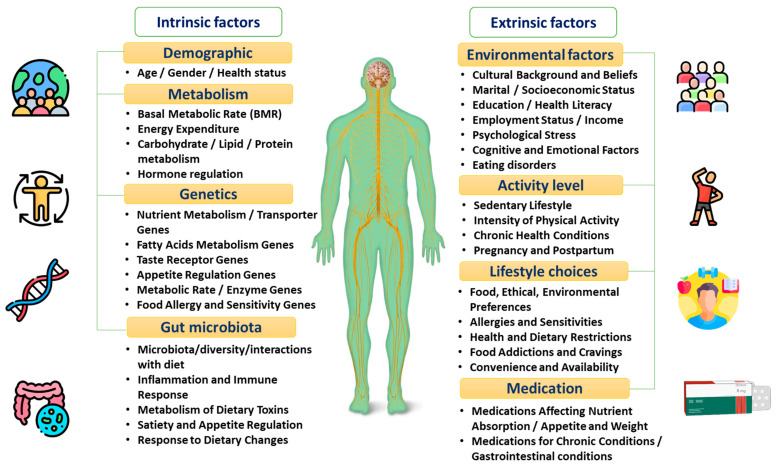
The factors affecting individuals’ dietary responses.

**Figure 2 nutrients-16-00581-f002:**
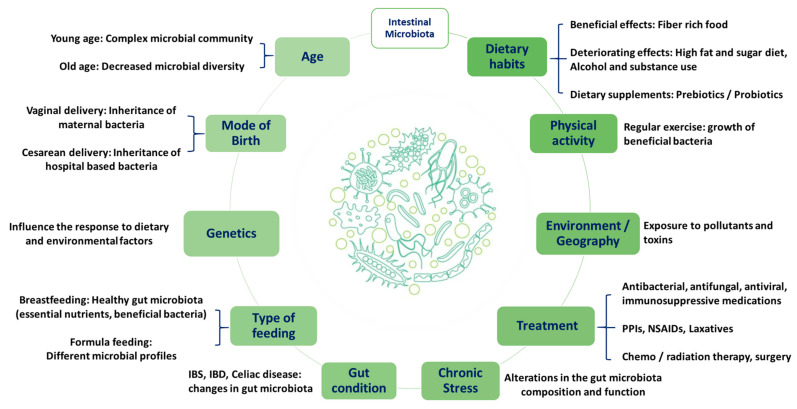
Factors affecting individuals’ intestinal microbiota. Proton pump inhibitors (PPIs), nonsteroidal anti-inflammatory drugs (NSAIDs), irritable bowel syndrome (IBS), and irritable bowel disease (IBD).

**Figure 3 nutrients-16-00581-f003:**
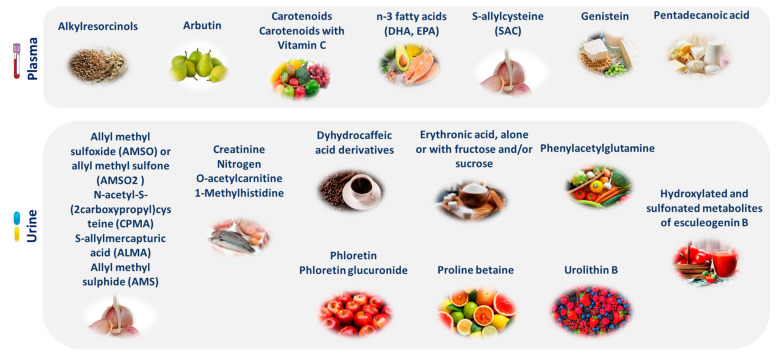
Evidence-based nutritional biomarkers in plasma and urine samples.

**Table 1 nutrients-16-00581-t001:** The role of genetic variants in weight management and metabolic phenotypes.

Type of Illness	Type of Intervention	*n*	Genetic Component	Key Finding	Ref.
Obesity	2 years of dietary intervention	322	Leptin (LEP) SNPs *	Possibility of regaining weight from 7 to 24 months	[81]
	2 years of dietary intervention	742	Fat mass and obesity-associated gene (FTO) SNP rs1558902	A high-protein diet in the presence of FTO genotype: (1) Remarkable weight loss (2) Amelioration in body composition and lipid distribution	[82]
	4 years of lifestyle-based intervention	3899	SNPs	Following weight loss, a risk of regaining weight was linked to FTO and BDNF loci	[83]
	2 years of lifestyle-based intervention and metformin use	3819	Melanocortin 4 Receptor gene (MC4R) SNPs	In the intervention group, the rs17066866 marker was associated with: (1) Less short-term weight loss (first 6-month period) (2) Less long-term weight loss (first 2-year period)	[84]
Diabetes	2 years of dietary intervention	738	Insulin receptor substrate 1 gene (IRS1) rs2943641	Altered intrinsic effect of dietary carbohydrate on weight loss and insulin resistance	[85]
	2 years of dietary intervention	591	Transcription Factor 7-Like 2 (TCF7L2) SNP rs7903146	Interaction between dietary fat intake and TCF7L2 🡪 modified BMI as well as total and trunk fat mass	[86]
	9 months of dietary intervention	304	TCF7L2 SNP rs7903146	Interaction between high-fibre dietary intake and CC genotype 🡪 improved weight loss	[87]
	2 years of dietary intervention	737	Glucose-dependent insulinotropic polypeptide receptor (GIPR) SNP rs2287019	Interaction of dietary carbohydrates with GIPR genotype 🡪 modifications in body weight, fasting glucose, and insulin resistance	[88,89]
	2 years of dietary intervention	738	IRS1 SNP rs1522813	Changes in dietary fat effects based on both IRS1 genotype and MetS status	[89,90]
Obesity, diabetes, or hypertension	4 months of dietary and medical intervention	722	21 SNPs	Interaction between dietary intervention and SNPs leads to changes in blood pressure	[91]
Hypertension	2 years of dietary intervention	723	Neuropeptide Y Promoter (NPY) SNP rs16147	Interaction between dietary fat and NPY genotype, modifying blood pressure	[92]

* Single-nucleotide polymorphism (SNP); *n*: sample size; Ref.: Reference.

## Data Availability

Not applicable.

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
