# Peer review of "Precision Nutrition Unveiled: Gene–Nutrient Interactions, Microbiota Dynamics, and Lifestyle Factors in Obesity Management"

_nutrients, 2024, doi:10.3390/nu16050581_

Round 1

Reviewer 1 Report

Comments and Suggestions for Authors

This paper reviews the existing literature on precision nutrition (PN) and its role in the management of obesity and other related conditions (specifically diabetes, cardiovascular disease, and cancer), identifying 3 main components of PN that are of interest, 1) gene-nutrient interactions, 2) intestinal microbiota, and 2) lifestyle factors. It posits that PN is a key strategy to address risk factors of obesity and concludes that PN is effective because of its ability to integrate genetic and lifestyle information to create individualized dietary recommendations.

The authors explore an important and interesting topic and have exhaustively reviewed over 200 papers. The effort to discuss the body of research on precision nutrition strategies is commendable and useful to reveal next steps in research and application. The authors are clearly trying to cover a lot of ground, but the aim/perspective gets lost without a clear outline. Additionally, the overall paper is difficult to read due to poor phrasing (e.g., lines 43-44), misuse of terms (e.g., “outdoor foods” (line 193); “low physical education” (line 290); “futuristic findings” (line 282)), overuse of concluding terms (e.g., therefore, thus, consequently, etc.) in unnecessary places (e.g., 4 times in the first paragraph of Introduction), and certain statements are out of place and do not follow logically from what was discussed immediately prior (e.g., lines 43-46, lines 280-282, lines 357-358). 

The paper would benefit significantly from a much-revised Introduction, which is currently dis-jointed and lacking a logical flow. It may be useful to lead with a discussion of the problem of obesity. Multiple factors contribute to obesity but lines 55-57 do not refer to the full scope and seem to imply that obesity and the other comorbidities are only due to a few factors. The last paragraph of the Introduction (lines 66-73) would seem more appropriate in the Conclusions section; and the first paragraph in Materials and Methods (lines 77-81) would make more sense in the Introduction. 

A Materials and Methods section may not be necessary in this review, but if including, it may be useful to address: How many papers were found; how many were used? What was criteria for inclusion or exclusion? Was there a focus or was every single paper found also cited? Maybe break down number of papers found per site or per keyword?

The main body discussing the research can be better synthesized. Certain sections are quite comprehensive (e.g., 3.1) while many others are not sufficiently treated (e.g., 3.3) and still others can be summarized more succinctly (e.g. 4.4). Based on the title and abstract, one would expect Section 3 to go into detail on the 3 main PN components that the authors point to in the order listed, but it does not. It may be useful to add an outline at the start of Section 3. 

It is important to note that not all nutrition is precision nutrition. What do the authors define as/understand to be “precision nutrition”? Some papers cited distinguish between “precision nutrition” versus “personalized nutrition” (Livingstone paper, citation 4) but authors do not make distinction.

Figures and Tables listed below need more explanation and discussion of relevance within the body of the manuscript
-       Figure 2; may be useful to discuss what intrinsic/extrinsic factors are, perhaps in Introduction or at the beginning of Section 3
-       Table 1; doesn’t seem to logically follow the section where it’s mentioned regarding weight management

I would recommend the authors to review the following paper as an example: Antwi, J. Precision Nutrition to Improve Risk Factors of Obesity and Type 2 Diabetes. Curr Nutr Rep 12, 679–694 (2023).

Comments on the Quality of English Language

see attached file

Author Response

The authors would like to thank the reviewer for allowing us to revise our manuscript. The authors also would like to thank the reviewer for the comments and recommendations that we believe have improved the quality of the manuscript. All amendments to the manuscript are made in blue font to facilitate your review.

Reviewer 2 Report

Comments and Suggestions for Authors

The authors describe several applications for precision nutrition, including biomarkers, genes, microbiota, and other health conditions. It is obvious how much work went into this paper, and I commend the authors. Because there is so much information and concepts, I recommend the authors narrow down the concepts so the paper is more focused.  My overall impression after reading it was that there were a lot of areas that PN could be applicable, but not a lot of PN study results were reviewed. I recommend focusing more on what PN studies have found, not just individual aspects that can be applied to PN models. 

Lines 53 - 55 state an obesity rate of 35% using old references. Please update this statistic and references. 

Lines 133 - 135 just add details about FODMAP and I don't see the connection to PN. Recommend adding more context or removing the sentence. 

Please follow scientific nomenclature for italicizing microbiota: https://wwwnc.cdc.gov/eid/page/scientific-nomenclature

Line 179-181 is confusing. Did research in these areas expand the ability to understand? 

The paragraph starting at line 285 mixes dietary PN and physical activity interventions. These two concepts seem separate and not related. Recommend reworking this paragraph to tie together the concepts or removing. 

Figure 3 needs more description of why those compounds were chosen or I recommend removing it. There are several biomarkers, and ones listed in the sentence referring to the figure (glucose, fats, etc.) that are not included. Why were these specific ones chosen, especially since they are not elaborated on in the text? 

Recommend remove section 4.4.3 Cancer. I don't think it adds value to the overall focus of the paper. Nutrition interventions can be used to improve cancer outcomes, but evidence of using PN for personalized interventions is not supported. 

In summary, the paper has too much information and the focus of PN is lost.

Comments on the Quality of English Language

The English language is appropriate. There are few words or phrases that are not used in their regular way, but not enough to require re-working. 

Author Response

(The authors gave the same response as above.)

Reviewer 3 Report

Comments and Suggestions for Authors

This manuscript summarized the recent study on Precision Nutrition (PN). Since diet is closely related to our health, in recent days, the importance of PN has attracted the attention of the healthcare field. This topic is timely and beneficial for people who have health issues, in particular obesity and associated conditions. The manuscript sections are good: Divided by Intrinsic factors and extrinsic factors. Also, socio-psychological variables are considered significant factors that affect their health. This manuscript focused on it and described how it impacts their lives.

The references were collected from the recent progress, mainly the past five years,  and the amount is enough. The contents are summarized from research and clinical aspects. The figures are supported by the text body.

The manuscript is valuable since it contains helpful information for further study and applications in this area.

I suggest publishing this manuscript to publish in Nutrients after these minor revisions to the format.

·         There are double periods.

4.4.1..

4.4.3..

·         The text in Figures 1-3 are very small. They are hard to read.  Please make them bigger. 

Author Response

(The authors gave the same response as above.)

Reviewer 4 Report

Comments and Suggestions for Authors

The authors further reviewed a recently recommended approach of “Precision Nutrition” to unveil interaction between personal gene variants, microbiota, and lifestyle in management of a complex disease of obesity, which would otherwise increase risk factor for T2D, cardiovascular disease, and cancer if it is not managed and body weight dramatically exceeds what is considered healthy for the patient’s height and BMI.

Although this topic and P.N. components, discussed in this review were previously reviewed, the authors in this current manuscript filled missing gabs/findings, used appropriate tools of search, presented coherent review with sound narration and illustrative schemes.  

However, the following comments and reviews in the scientific drafting as indicated below still needs to be addressed, re-drafted, revised and rectified for more readable, sound, and informative published text, and more critical review in these particular sentences.

1.       In table 1 (page 8), the last two SNPs towards the bottom of the table: SNP rs2287019 and SNP rs1522813 (referenced by ref. 135, 136, respectively): Since these two SNPs were reviewed by Yoriko Heianza  and Lu Qi 2017, International J of Molecular Sciences 18, 787; doi:10.3390/ijms18040787 (MDPI-published), in table 2 page 4 of that review, it would be appropriate to cite that paper beside refence 135, 136 in table 1 page 8 of this current manuscript given that description in table legend in both manuscripts are scientifically synonyms. Any other significant critical comment in that paper can also be incorporated and cited if it adds to any context in the current manuscript.

2.       In page 10 line 367-369 the sentence: The intestinal microflora also differs in diabetic patients with lower level of butyrate-producing bacteria compared to healthy patients.

2.1. The above stated sentence does not read well. It should be: “with lower level of butyrate-producing bacteria in diabetic patients compared to healthy patients/subjects.

2.2.  As a critical review, to say butyrate-producing bacteria is not a favorable and not up to date as Larsen, N. et al. 2010 PLoS One, 5e, e9085 (ref. no. 168 in the current manuscript) reported that the ratios of Bacteroidetes to Firmicutes correlates positively with plasma glucose concentrations. If the ratio is higher in diabetics (as per the positive correlation with glucose level) than healthy, this means less Firmicutes in diabetics, where Firmicutes are butyrate-producing bacteria. Given this cited paper was reported in 2010, more recently Vital M. et a. 2014, Apr 22;5(2): e00889. doi: 10.1128/mBio.00889-14 reported that Bacteroides produce butyrate. Then it is simpler to state that the ratio of Bacteroides to Firmicutes correlated positively with plasma glucose level in, “” with lower Firmicutes in diabetic patients as compared the non-diabetic healthy controls””.

2.3. As per the required critical review beyond the sole narration of the cited results: it fair to add: further confirmation by causal, quantitative and/or appropriate mechanistic analysis still worth pursing to confirm this correlation and association phenomena.

2.4. The reference no. 169 (Lambeth S.M. et al. 2015 J Diabetes Obes 2015 Dec 26;2(3):1-7) in this manuscript can be removed or cited in separate sentence to report difference in microbiota between preDM compared to T2DM but the authors did not find relationship between microbiota composition and diagnostic group or HbA1c as reviewed by 

 3.    In the same page (10), line 369-371, pasted from the text “ More specifically, Prevotellacopri (P. Copri) and Bacteriodes vulgatus species were seen to stimulate insulin resistance by activating the branched-chain amino acids (BCAAs) biosynthesis. It is more informative to add: “which in turn signal to mTOR complex leading to phosphorylation of p70S6 serine Kinase (S6K1), which inactivates insulin receptor substrate thereby promoting insulin resistance by Goda, J. and Cahova, M. 2021 Biomolecules 2021, 11, 1414. https://doi.org/10.3390/biom11101414.

Alternatively, it can be drafted as “for further details on the link between the elevated serum BCAA and insulin resistance see/visit Goda J. and Cahova M. 2021 Biomolecules 2021, 11, 1414. https://doi.org/10.3390/biom11101414  or as narrated and reviewed elsewhere (Ref. Goda J. and Cahova M. 2021 Biomolecules 2021)

This review paper in Molecules 2021 should be cited as it clearly and properly explains the role of high serum BCAA as a metabolite affected by specific bacteria and cause insulin resistance, further supporting the statement

Comments on the Quality of English Language

The manuscript is written in perfect, sound and clearly readable English language. The suggested reviews and comments are scientific in nature.

It is evident that the text language was reviewed and edited by professional/native speaking editors.

Author Response

(The authors gave the same response as above.)

Round 2

Reviewer 1 Report

Comments and Suggestions for Authors

I appreciate the extensive re writing and editing. the manuscritp is improved. There is still slang and causual use of language that detracts from the message that could be improved with an editor. 

Comments on the Quality of English Language

still needs extensive editng

Reviewer 2 Report

Comments and Suggestions for Authors

Thank you for revising the manuscript to tie the various points together. It really elevated the paper and relevance of the different topics. I also appreciate the modification of the figures.